# Blood Coagulation and Thrombotic Disorders following SARS-CoV-2 Infection and COVID-19 Vaccination

**DOI:** 10.3390/biomedicines11102813

**Published:** 2023-10-17

**Authors:** Metodija Sekulovski, Niya Mileva, Georgi Vasilev Vasilev, Dimitrina Miteva, Milena Gulinac, Monika Peshevska-Sekulovska, Lyubomir Chervenkov, Hristiana Batselova, Georgi Hristov Vasilev, Latchezar Tomov, Snezhina Lazova, Dobrin Vassilev, Tsvetelina Velikova

**Affiliations:** 1Department of Anesthesiology and Intensive Care, University Hospital Lozenetz, Kozyak Str., 1407 Sofia, Bulgaria; 2Medical Faculty, Sofia University, St. Kliment Ohridski, Kozyak 1 Str., 1407 Sofia, Bulgaria; vvasilev.georgi@gmail.com (G.V.V.); d.georgieva@biofac.uni-sofia.bg (D.M.); mgulinac@hotmail.com (M.G.); mpesevska93@gmail.com (M.P.-S.); drgeorgivasilev@gmail.com (G.H.V.); lptomov@nbu.bg (L.T.); snejina@lazova.com (S.L.); tsvelikova@medfac.mu-sofia.bg (T.V.); 3Medical Faculty, Medical University of Sofia, 1 Georgi Sofiiski Str., 1431 Sofia, Bulgaria; nmileva91@gmail.com; 4Clinic of Endocrinology and Metabolic Disorders, University Multiprofil Hospital Active Treatement “Sv. Georgi”, 4000 Plovdiv, Bulgaria; 5Department of Genetics, Faculty of Biology, Sofia University “St. Kliment Ohridski”, 8 Dragan Tzankov Str., 1164 Sofia, Bulgaria; 6Department of General and Clinical Pathology, Medical University of Plovdiv, Bul. Vasil Aprilov 15A, 4000 Plovdiv, Bulgaria; 7Department of Gastroenterology, University Hospital Lozenetz, 1407 Sofia, Bulgaria; 8Department of Diagnostic Imaging, Medical University of Plovdiv, Bul. Vasil Aprilov 15A, 4000 Plovdiv, Bulgaria; lyubo.ch@gmail.com; 9Department of Epidemiology and Disaster Medicine, Medical University of Plovdiv, University Hospital “St George”, 4000 Plovdiv, Bulgaria; dr_batselova@abv.bg; 10Laboratory of Hematopathology and Immunology, National Specialized Hospital for Active Treatment of Hematological Diseases, 1756 Sofia, Bulgaria; 11Department of Informatics, New Bulgarian University, Montevideo 21 Str., 1618 Sofia, Bulgaria; 12Pediatric Clinic, University Hospital “N. I. Pirogov”, 21 “General Eduard I. Totleben” Blvd, 1606 Sofia, Bulgaria; 13Department of Healthcare, Faculty of Public Health “Prof. Tsekomir Vodenicharov, MD, DSc”, Medical University of Sofia, Bialo More 8 Str., 1527 Sofia, Bulgaria; 14Faculty of Public Health and Healthcare, Ruse University Angel Kanchev, 7017 Ruse, Bulgaria; dobrinv@gmail.com

**Keywords:** blood disorders, blood coagulation, thrombophilia, hemophilia, thrombotic incidents, thrombotic events, cerebral venous thrombosis, thrombocytopenia, disseminated intravascular coagulation, COVID-19, COVID-19 vaccines

## Abstract

Although abundant data confirm the efficacy and safety profile of the developed vaccines against COVID-19, there are still some concerns regarding vaccination in high-risk populations. This is especially valid for patients susceptible to thrombotic or bleeding events and hesitant people due to the fear of thrombotic incidents following vaccination. This narrative review focuses on various inherited and acquired thrombotic and coagulation disorders and the possible pathophysiologic mechanisms interacting with the coagulation system during immunization in view of the currently available safety data regarding COVID-19 vaccines. Inherited blood coagulation disorders and inherited thrombotic disorders in the light of COVID-19, as well as blood coagulation and thrombotic disorders and bleeding complications following COVID-19 vaccines, along with the possible pathogenesis hypotheses, therapeutic interventions, and imaging for diagnosing are discussed in detail. Lastly, the lack of causality between the bleeding and thrombotic events and COVID-19 vaccines is debated, but still emphasizes the importance of vaccination against COVID-19, outweighing the minimal risk of potential rare adverse events associated with coagulation.

## 1. Introduction

The COVID-19 pandemic caused by severe acute respiratory syndrome coronavirus-2 (SARS-CoV-2) has affected hundreds of millions worldwide, leading to nearly 7 million deaths globally, although now declared not a worldwide concern anymore [1]. Strenuous research and analysis of various vaccine advances led to the development of multiple COVID-19 vaccines in less than a year from the pandemic’s beginning. Different types of vaccines, such as mRNA vaccines, DNA vaccines, viral vector vaccines, and inactivated virus vaccines have been approved and have shown a high degree of efficacy with variable protective levels of up to 95% (70–95% range) in vaccinated individuals against COVID-19 [2,3,4,5,6].

Despite the abundance of data revealing the safety and efficacy of the developed vaccines [5,7], there are still some concerns regarding the challenges related to vaccination in high-risk populations, especially in patients susceptible to thrombotic or bleeding events.

This narrative review focuses on various inherited thrombotic and coagulation disorders (inherited blood coagulation disorders and inherited thrombotic disorders in the light of COVID-19), and acquired thrombotic and coagulation disorders (blood coagulation and thrombotic disorders and bleeding complications following COVID-19 and COVID-19 vaccines), along with the possible pathogenesis hypotheses, therapeutic interventions, and imaging for diagnosing. We discuss possible pathophysiologic mechanisms interacting with the coagulation system during vaccination and review the currently available safety data regarding COVID-19 vaccines.

## 2. Search Strategy

For our narrative review, we extensively searched the scientific databases Medline (PubMed) and Scopus, along with strong supporting data from official registries. We used the following MeSH and pertinent free-text terms: (“COVID-19” OR “SARS-CoV-2” AND (“blood disorders” OR “coagulation disorders” OR “thrombotic event”) AND (“vaccine” OR “mRNA” OR “vector” OR “inactivated”). We only looked for items released before 10 July 2023. Our review structure was based on the guidelines and recommendations for writing a narrative review [8].

## 3. SARS-CoV-2 and COVID-19 Vaccines

Different types of vaccines against the SARS-CoV-2 virus were developed to prevent the subsequent COVID-19. Vaccines can be divided into the following categories based on the platform used for their development: DNA, mRNA, inactivated, live attenuated, virion-like particle, viral vector, recombinant subunit and synthetic peptide (Figure 1). The figure presents the most severe conditions reported as adverse events following vaccination, with 50 already approved COVID-19 vaccines as of the end of 2022 [9]. 

The data on the severe adverse effects of COVID-19 vaccines should be interpreted carefully considering the CDC, ACIP, and WHO conclusion that COVID-19 vaccination outweighs any potential risks of complications, based on the most intense safety monitoring program in the U.S. history and worldwide [10]. 

Some of the COVID-19 vaccines are currently available worldwide [11]. We will briefly introduce them, along with some concerns regarding their use.

One of these vaccines uses synthetic mRNA, formulated in vitro and introduced into the human body for translation into antigenic proteins by the host cells [12,13]. The bioproduct BNT162b2 (Pfizer/BioNTech) is a messenger RNA vaccine and contains a lipid nanoparticle. Nucleoside-modified RNA (modRNA) encodes the full-length spike protein of SARS-CoV-2, modified by two proline mutations to lock it into a preconfirmation [14,15,16]. mRNA-1273 is a vaccine developed by Moderna and consists of a messenger RNA that encodes the spike protein stabilized in a preconformation, or SARS-CoV-2 S(2P), formulated in lipid nanoparticles [17].

On the other hand, viral vector vaccines aim to insert a gene encoding protective exogenous antigens into a viral vector to express the target protein in the human body [12,13]. ChAdOx1-S (Oxford-AstraZeneca) is a vaccine containing the adenovirus family virus modified to include a protein-making gene from SARS-CoV-2 [18]. The adenovirus is a recombinant chimpanzee vector encoding the spike glycoprotein of SARS-CoV-2 [19]. Ad26.COV2.S (Janssen) is a bioproduct containing a recombinant, replication-incompetent human adenovirus type 26 vector encoding the spike protein of the SARS-CoV-2 virus in a conformation stabilized by pre-fusion for prevention of COVID-19 [20]. Ad5-nCoV (Convidecia™) is a vaccine containing a non-replicating viral vector adenovirus Type 5 vector, developed by CanSino Biologics/Beijing Institute of Biotechnology/Petrovax [21,22].

Sputnik V, or Gam-Covid-Vac, is a recombinant adenovirus vector vaccine by Gamaleya Research Institute [23,24]. The first dose of the bioproduct contains recombinant adenovirus rAd26, and the second dose adenovirus rAd5. Both vectors carry the gene for the full-length SARS-CoV-2 glycoprotein S [23,24].

In Turkmenistan, a synthetic peptide vaccine was approved to prevent COVID-19. It was developed by Vector Institute and named EpiVacCorona [25]. Inactivated vaccines against COVID-19 contain whole-virus cultured wild-type viruses by physical or chemical inactivation processes [12,13]. One bioproduct with an inactivated virus for the prevention of COVID-19 is BBIBP-CorVApproved, developed by the Beijing Institute of Biological Products, and approved for use in Bahrain, China, and the United Arab Emirates [26,27]. 

Wuhan Institute of Biological Products/Sinopharm produced another inactivated vaccine against the SARS-CoV-2 Vero cell [26]. In Russia, the inactivated CoviVac vaccine was developed by the Chumakov Center at the Russian Academy of Sciences [28]. Another type of vaccine is Bionic nanoparticle vaccines, which are bioproducts with purified recombinant proteins and bionic nanoparticles [12,13]. NVX-CoV2373 is a recombinant nanoparticle vaccine composed of trimeric full-length SARS-CoV-2 spike glycoproteins. The bioproduct contains Matrix-M1 adjuvant [29].

These approved vaccines underwent firm clinical trials and were investigated during real-life use, while minor and major adverse events associated with vaccines were strictly registered and monitored [10].

## 4. Inherited Blood Coagulation Disorders and COVID-19 Vaccines

Coagulopathic hemorrhagic diatheses (coagulopathies) are conditions caused by a congenital or acquired defect in the blood coagulation system. Hemorrhagic diathesis represents an increased tendency to bleed. Hereditary bleeding disorders may occur due to defective platelet function, and, depending upon the predominant functional abnormality, inherited disorders are classified into the three most common groups: defective platelet adhesion, defective platelet aggregation, and disorders of platelet release reaction [30,31]. 

Most inherited coagulation disorders are induced by qualitative and quantitative defects in a single coagulation factor. Two of the most common factors that are reported are the sex (X)-linked disorders—classic hemophilia or hemophilia A (factor VIII deficiency) and hemophilia B or Christmas disease (factor IX deficiency). Another common and related coagulation disorder is von Willebrand’s disease (defect of von Willebrand’s factor). 

Hemophilia A is the second most common coagulation disorder, next to von Willebrand’s disease. It is inherited in an X-linked recessive pattern. Therefore, females are carriers, whereas the disease manifests clinically in males. The frequency of this type of disorder varies in different races, with the highest incidence being in British populations, particularly in royal blood descendants and some European royal families. These patients suffer from bleeding involving any organ for hours or days after injury, and the severity of bleeding correlates with plasma level factor VIII activity. Broadly, the most severe, non-lethal changes in Hemophilia A result from intra-articular hemorrhages that lead to deforming arthrosis. On the other hand, even minor traumas, such as tooth extraction, can lead to massive, life-threatening bleeding in these subsets of patients [32,33,34].

Since COVID-19 is associated with coagulation and thrombotic disorders, little was known about the infection’s impact on the clinical outcomes of patients with hemophilia. A retrospective study by Mericliler et Narayan with 1758 adult male patients reported that COVID-19 did not increase the mortality of COVID-19 in hemophilia A patients but increased the risk of bleeding and hospitalizations [35]. 

WHO and The World Federation of Hemophilia (WFH) guidelines recommend that patients with hemophilia preferably receive subcutaneous vaccination; the problem comes from the fact that the vaccine for COVID-19, as well as most vaccines, is only allowed for intramuscular administration. Bleeding following the administration of an intramuscular vaccine in hemophiliacs is challenging to control. It may require several days, even weeks, of treatment with a clotting factor [36]. These precautions are also valid for patients on anticoagulants or antiplatelets [37].

However, there are some reports of patients with acquired hemophilia A following COVID-19 vaccination [38,39,40,41,42,43,44]. Duminuco et al. described this rare coagulopathy associated with hemorrhagic complications due to the possible development of FVIII inhibitors following an immune stimulus [38].

Von Willebrand’s disease (Pseudohemophilia) has an autosomal dominant inheritance. In this disease, impaired platelet aggregation is detected. Its incidence is estimated to be 1 in 1000 of either sex. Clinically, the patients are characterized by spontaneous bleeding from mucous membranes and excessive wound bleeding [45,46].

As expected, the von Willebrand factor plays a role in COVID-19-associated coagulopathy, as its levels and activity increase in severe infection [47]. However, little to no information is available on COVID-19’s impact on patients with this disease [48]. 

Hemophilia B (Christmas disease) is rarer than hemophilia A, with clinical features indistinguishable from classic hemophilia [49]. Not much is known about COVID-19 in these patients. However, managing underlying hemophilia of any type requires hemostatic treatment for ambulatory patients and prevention with concentrates intensified according to the risk of complications in hospitalized patients and those on replacement therapy [50]. 

## 5. Congenital Coagulation and Thrombotic Disorders and COVID-19 Vaccines

Thrombophilia is a type of hypercoagulability that represents a pathologic state of increased clot formation without active bleeding [51]. 

Factor V Leiden mutation is the most common inherited thrombophilia in individuals with venous thromboembolism. The mutation leads to enhanced pro-thrombotic actions of activated factor Va by changing the cleavage site of the molecule and hindering its degradation by protein C. It is an autosomal dominant genetic condition with incomplete penetrance, so not every carrier of the mutated gene will exhibit a thrombotic event [52]. 

Protein C and S deficiency: Protein C is a vitamin K-dependent protease presented in low levels in human plasma. Once activated, it cleaves the activated factors V and VIII, thus inhibiting thrombin production and coagulation. The functions of protein C in inflammation and cytoprotection have been described unrelated to its role in coagulation. Protein S is a cofactor to activate protein C in the cleavage of Va and VIIIa, as well as a cofactor of the tissue factor pathway inhibitor (TFPI) [53]. Protein C and S deficiencies may be inherited and acquired. There are two types of inherited forms, most commonly with autosomal recessive inheritance. Type I is associated with decreased levels and activity of the proteins, while type II is related to normal levels but reduced anticoagulant activity. Both protein C and S deficiencies are associated with uncontrolled thrombin formation and thromboembolic events, with venous thromboembolism being more common than arterial [54]. 

Antithrombin III deficiency: Antithrombin III is a glycoprotein located on the vascular surface of endothelial cells that requires heparin as a cofactor with which they bind circulating thrombin, thus inhibiting coagulation. Antithrombin III deficiency is inherited in an autosomal dominant manner and presents almost exclusively with venous thrombosis. The risk of thrombotic events is higher than in protein S and C deficiencies. Antithrombin III deficiency is estimated to carry the highest venous thromboembolism risk among all hereditary thrombophilias [55].

Hyperhomocysteinemia: Hyperhomocysteinemia has been notorious for decades for causing widespread accelerated atherosclerosis and arterial thrombosis. Homocysteine is an amino acid produced in methionine metabolism that causes endothelial injury and oxidative stress when present in substantial amounts. For homocysteine to be resynthesized to methionine by the enzyme methionine synthase, a methyl group is needed in the form of methyl-tetrahydrofolate delivered by the methylenetetrahydrofolate reductase (MTHFR). Several polymorphisms of the MTHFR gene are associated with significantly elevated serum homocysteine levels, with the homozygous mutant genotype leading to the greatest homocysteine values [56]. 

A large observational study from the Mayo Clinic, Rochester, Minnesota, USA, identified 6067 patients with confirmed thrombophilia among almost 800,000 vaccinated patients against COVID-19. Subgroup analysis did not find any statistically significant difference in the occurrence of venous thromboembolism between patients with and without thrombophilia vaccinated against COVID-19 [57]. 

It has even been estimated that the absolute risk for a deep vein thrombosis while undergoing an airplane flight is 1 in 4600 people, making it 50 to 100 times more likely than thrombosis following SARS-CoV-2 vaccination [58]. A European expert consensus on Vaccine-Induced Immune Thrombotic Thrombocytopenia recommends against the systematic screening for thrombophilia; against premedication with low-molecular-weight heparin, direct oral anticoagulants, or aspirin; as well as against monitoring of changes in D-dimer [58]. 

## 6. Review of Acquired Coagulation and Thrombotic Disorders and Their Connection with COVID-19 Vaccines—Hypothesis and Therapeutic Interventions

As we previously described in detail, acquired coagulation disorders are a heterogeneous group of conditions with a complex pathophysiology. The major causes of these disorders include the development of circulating anticoagulants, impaired synthesis of clotting factors, and the development of disseminated intravascular coagulation (DIC) [59].

Circulating anticoagulants are usually autoantibodies that “attack” specific clotting factors, such as an autoantibody against factor VIII or factor V, or inhibit phospholipid-bound proteins. Patients may develop lupus anticoagulant hypoprothrombinemia or antiphospholipid syndrome. Occasionally, the latter type of autoantibody causes bleeding by binding to prothrombin–phospholipid complexes [60]. Such antibodies may develop due to autoimmune disease or be drug-induced. There are few case reports (documented patients who developed acquired hemophilia following COVID-19 vaccination.)

On the other hand, synthesis of clotting factors may be impaired either by severe liver disease (such as fulminant hepatitis, cirrhosis, acute liver failure) or by deficiency of vitamin K that is required for the synthesis of prothrombin, factor VII, factor IX, protein C, and protein S [61]. Vitamin K deficiency usually results from deficient intake or conditions causing fat malabsorption, such as coeliac disease or cystic fibrosis. Furthermore, acute liver disease may cause disproportionate fibrinolysis and bleeding due to decreased synthesis of antiplasmin. To the best of our knowledge, no known reported cases are associated with impaired clotting factor synthesis and COVID-19 vaccination.

Disseminated intravascular coagulation involves a disproportionate synthesis of thrombin and fibrin in the circulating blood, and it usually develops from exposure of tissue factor to blood, initiating the extrinsic coagulation cascade [62]. Cytokine release and impaired microvascular blood flow provoke tissue plasminogen activator (tPA) release from endothelial cells. Both tPA and plasminogen attach to fibrin, and plasmin (generated by tPA cleavage of plasminogen) divides fibrin into D-dimers and other fibrin degradation products. Therefore, excessive platelet aggregation and coagulation factor consumption occur. Therefore, DIC may cause both thrombosis and bleeding. Furthermore, administering anticoagulants may lead to bleeding events because of DIC evolution; thus, this scenario must be considered.

On the other hand, DIC that progresses briskly (hours or days) usually causes bleeding events. DIC usually presents with thrombocytopenia, increased prothrombin time and plasma D-dimer levels, and decreased fibrinogen levels. Several cases of DIC in patients have been described after adenovirus vector vaccination (ChAdOx1-nCoV-19 and Ad26.COV2.S) in the literature [63,64].

## 7. Blood Coagulation and Thrombotic Disorders Related to COVID-19 Disease

The variety of clinical symptoms associated with novel coronavirus infection astounded researchers. The condition can be asymptomatic, with only modest signs such as olfactory loss, overall weakness, or flu-like symptoms. However, COVID-19 infection can be severe in some patients, resulting in hypercoagulation, vascular endothelial damage, and the risk of venous and arterial thrombotic consequences [65]. Multiple research publications covering various features and signs of the disease have been published in recent months. They reported that the endothelial cells in COVID-19 play a pivotal role as mediators of the two-way communication between inflammation and coagulation [66]. On one hand, SARS-CoV-2 binding to the endothelial angiotensin-converting enzyme 2 receptor activates endothelial cells through a complicated, inflammatory response [67]. 

On the other hand, studies using COVID-19-infected lung tissue have also shown enhanced expression of vascular and inflammatory factors. These factors include vascular cell adhesion molecule (VCAM)-1, interleukin (IL)-8, and monocyte-chemoattractant protein (MCP)-1. Furthermore, scientific research has proven that platelet adhesion molecule vWF is upregulated in COVID-19, demonstrating endothelial involvement; however, a disintegrin and metalloproteinase with thrombospondin motifs 13 (ADAMTS13), which cleaves high-molecular-weight vWF, is downregulated, resulting in an aberrant ratio of vWF to ADAMTS-13 [68]. This disturbed ratio is at the base of the pathogenesis of COVID-19-associated coagulopathy.

In line with this, it is unsurprising that severe disease and subsequent cardiovascular consequences after SARS-CoV-2 infection are higher in patients with preexisting vascular disease, such as hypertension, diabetes, and coronary artery disease. Since the lungs are the entryway for most viruses and SARS-CoV-2, the pulmonary vasculature is the first to experience inflammation. Barrier function and vascular tone may also be indirectly affected by microvascular thrombosis. Vascular dysfunction and microvascular thrombosis contribute significantly to further developing pulmonary illness, particularly ARDS. There have been reports of increased thrombotic events such as venous thromboembolism, myocardial infarction, and stroke [69]. 

Although SARS-CoV-2 infection, which causes coronavirus illness (COVID-19), has been linked to thrombotic problems in adults, the prevalence of COVID-19-related thrombosis in children and adolescents is unknown. Mild disease is typical for children with acute COVID-19, but the postinfectious complication known as multisystem inflammatory syndrome in children (MIS-C) has mostly been linked to coagulopathy [70]. Evidence demonstrated that MIS-C and other post-COVID-19 complications may originate in vascular dysfunction. For example, since April 2020, previously healthy children have presented with fever, cardiovascular shock and/or Kawasaki disease symptoms, hyperinflammation, and multisystem involvement after SARS-CoV-2 infection [71]. Many of them had positive SARS-CoV-2 antibody titers but negative nasopharyngeal swabs. Thus, CDC and WHO public health advisories listed criteria for this new disease, a multisystem inflammatory syndrome in children (MIS-C) associated with coagulopathy, as a potential presenting feature [71,72,73]. 

Therefore, it is essential to underline that immobile patients with severe infection and those with severe inflammatory reactions have an elevated risk of VTE, a key issue for all hospitalized patients. Consequently, preventative measures, including pharmaceuticals and mechanical devices, should be employed, and early mobilization should be encouraged. Since pharmaceutical thromboprophylaxis has been demonstrated to benefit hospitalized patients with COVID-19, a higher dose of anticoagulation may be necessary to treat the severe coagulopathy associated with this virus. In an effort to strike a better balance between thrombotic and bleeding events, it was proposed early on in the pandemic that the dose of thromboprophylaxis be established to empirical therapeutic-dose anticoagulation or to intermediate-dose anticoagulation [74]. Thus, only thromboembolic problems warrant therapeutic-dose heparin in critically ill COVID-19 patients. All patients can prevent VTE with LMWH in prophylactic doses. Some ICU specialists favor unfractionated heparin infusion for VTE in renal failure (ideally guided by anti-factor Xa levels since the aPTT is inaccurate in COVID-19 patients due to excessive factor VIII levels). Heparin-induced thrombocytopenia patients could benefit from fondaparinux, argatroban, or bivalirudin [75]. However, with the growing incidence of bleeding reports associated with COVID-19, several articles recommend using anticoagulation with serious percussion and individually assessing the benefit/risk ratio because of their adverse effects (i.e., paradoxical bleeding during DIC evolution).

With respect to the prognosis, important indicators of poor outcomes in SARS-CoV-2 infections include markers of endothelial dysfunction and altered endothelial cell integrity, which are linked to pulmonary edema, intravascular thrombosis, and acute respiratory distress syndrome (ARDS). The pulmonary endothelium is critical in maintaining vascular homeostasis and permeability, regulating inflammatory responses, coagulation, and fibrinolysis [76]. Disturbances in these tightly controlled processes may directly lead to morbidity and death.

## 8. Thrombotic and Bleeding Events following COVID-19 Vaccination: Hypothesis for Pathogenesis and Therapeutic Interventions

With the approval of the COVID-19 vaccines, the rate of hospitalizations and severe COVID-19 disease has decreased dramatically. The clinical trials of these vaccines reported a safety profile with the reactogenicity; adverse effects such as pain at the site of inoculation, redness, fever, fatigue, headache, and allergic reactions, have been reported [5,77,78,79]. However, at the beginning of 2021, the first cases of vaccine-induced thrombotic complications after vaccination with vector vaccines ChAdOx1 nCoV-19 (AstraZeneca) and Ad26.COV2.S (Johnson & Johnson/Janssen) were observed soon after all global vaccination programs were initiated. Subsequently, the relationship between thrombocytopenia and thrombosis after administration of BNT162b2 (Pfizer-BioNTech) and mRNA-1273 (Moderna) was also reported.

Different terms are used to refer to these conditions/complications after COVID-19 vaccination, which are predominantly localized in the venous vasculature system—vaccine-induced thrombotic thrombocytopenia (VITT), vaccine-induced pro-thrombotic immune thrombocytopenia (VIPIT), vaccine-induced immune thrombotic thrombocytopenia (VIITT) and thrombosis with thrombocytopenia syndrome (TTS) [80]. Antibodies that recognize PF4 (platelet-associated factor 4) cause these thrombotic events. The antibodies are IgG that activate platelets through specific receptors on the platelet surface. These events develop between 4 and 42 days after receiving a COVID-19 vaccine [81,82]. Figure 2 illustrates a suggested diagnostic algorithm in patients with suspected VITT. 

The advent of the COVID-19 pandemic has necessitated the development of vaccines as a crucial intervention against this highly aggressive and contagious viral disease. Multiple manufacturing companies employed various techniques, including viral nucleic acids, inactivated viruses, or viral vectors, to develop vaccines to mitigate the impact of this worldwide pandemic. However, the other side of the coin is vaccine adverse reactions. Among the most serious ones is thrombocytopenia, which has been shown to be due to both COVID-19 disease and immunization. When adenoviral-based vaccines were invented and administered, reported cases of patients with thrombosis and thrombocytopenia were later found to have VITT. Venous thromboses in unexpected places, such as cerebral sinus vein thrombosis or splanchnic vein thrombosis, have been observed in COVID-19 vaccine recipients, occasionally with thrombocytopenia (thrombosis thrombocytopenia syndrome [TTS]) [83]. Vector-based COVID-19 vaccination increased TTS-related cerebral sinus vein thrombosis risks by 50-fold. These findings support a pathogenetic relationship between vector-based COVID-19 vaccinations and TTS. Vaccine antigenic complexes bind to platelet factor 4 (PF4) on platelet surfaces, causing proinflammatory responses, anti-PF4 antibody production, and pro-thrombotic cascades [84]. Similarly, VITT is characterized by the production of platelet-activating antibodies targeting PF4. This clinical manifestation bears a resemblance to autoimmune heparin-induced thrombocytopenia [85]. 

Non-vector-based immunization was also related to cerebral thrombosis but without TTS. Sharifian-Dorche et al. conducted a study that revealed that nearly all of the patients with VITT exhibited heightened levels of antibodies to platelet factor 4 heparin complex, which is comparable to the occurrence of “classic” heparin-induced thrombocytopenia (HIT) [86]. But opposite to “classic” HIT, autoimmune heparin-induced thrombocytopenia causes severe thrombocytopenia, increased disseminated intravascular coagulation, and atypical thrombotic events.

TTP has traditionally been characterized by microangiopathic hemolytic anemia, thrombocytopenia, neurological abnormalities, fever, and renal failure [87]. From 5% in Australia to 17–55% in the United Kingdom (UK), VITT is a potentially major adverse event with a high mortality rate [88,89,90]. 

By the end of September 2021, 419 cases had been documented in the UK, and 89% of those cases started after the initial dose [20]. According to our data, the global fatality rate is 51.3%, and it is higher in situations involving the central nervous system, bleeding, a low platelet count, and elevated D-dimer levels. (Table 1) A recent review found that following ChAdOx1 nCoV-19 (Oxford/AstraZeneca) immunization, VITT overall mortality was 35.9%. The occurrence of intracerebral hemorrhage (ICH), age 60 years, platelet count 25 103/L, fibrinogen 150 mg/dL, and CVST were found to be strongly linked with death and were chosen as predictors for mortality [91]. TTP incidents following the COVID-19 vaccine have been more common in women under 55 and occur 5–30 days after immunization with virus-vectored vaccines (i.e., ChAdOx1 nCoV-19, Astra-Zeneca [Cambridge, UK] and Ad26.COV2.S Janssen [Beerse, Belgium]) [92]. The proposed therapeutic strategy included plasma exchange and corticosteroid therapy with an excellent effect [93]. Concerning anticoagulants, medication selection demands serious thought. Heparin is strictly prohibited in patients with classical HIT. However, Tiede et al. reported heparin therapy on one of their TTS patients. They found no indication that heparin was promoting platelet aggregation [94]. Still, caution is needed since, even if anti-PF4 antibodies were not brought on by heparin, the polyanion heparin may increase the possibility of aggregation in some subjects. The alternative drug argatroban seems to be both secure and efficient.

Despite rare occurrences of thrombosis with thrombocytopenia syndrome, COVID-19 vaccinations have a favorable benefit/risk profile. Table 1 presents the VITT incidents following COVID-19 vaccination based on the data in the literature [85,86,95,96,97,98,99,100].

## 9. Summary of Reported Literature Cases of Ultrarare VITT Complications

Table 1 summarizes the VITT reported incidents after the COVID-19 vaccine administration. Based on the data in the literature, we can conclude that most of the reported cases were following vector vaccines. 

Cerebral venous thrombosis (CVT) is another complication that is more serious than VITT. It includes thrombosis in the veins of the cranial sinuses, cortical veins and deep venous structures [101]. The CVT incidence in the general population is estimated at 2 cases per 100,000 persons per year [102,103]. EMA data show that CVT after vaccination administration occurs most frequently in women under 60 [104,105].

The occurrence of potentially life-threatening complications after COVID-19 vaccination with adenoviral vector vaccines prompted temporary suspensions of these vaccines. In April 2021, vaccinations with Ad26.COV2.S were temporarily suspended first in the USA due to the reporting of more than ten cases of CVT and thrombocytopenia following administration of Ad26.COV2.S vaccine [106]. Subsequently, several more studies reported cases of VITT patients after vaccination [80,81,107]. The data showed that 48 of 52 patients tested for anti-PF4 antibody had a positive result. In addition, the studies showed that women were three times more likely than men to acquire VITT, and over 88.5% of patients were under 60 years old [108]. Other studies also have shown that these adverse effects were observed more often in women in women between 40 and 60 years [109,110,111]. The known risk factors, for now, are pregnancy, postpartum period, use of oral contraceptive pills, and hormone therapy [112,113].

A study conducted by Greinacher et al. also reported 11 patients with thrombosis or thrombocytopenia after administration of the AstraZeneca vector vaccine, ChAdOx1nCov-19 [73]. Of them, nine were women under 50 years old. Symptoms appeared between 5 and 16 days after administering the first vaccine dose. All patients had severe or moderate thrombocytopenia and the presence of anti-PF4 antibodies. In addition, except for the anticoagulant treatment of these patients, high doses of immunoglobulins (IVIG) were administered to interrupt the pro-thrombotic VITT mechanism and inhibit platelets. Many other reports of thrombotic events after the AstraZeneca vaccination have followed [63,92,111,114,115,116,117,118,119,120,121,122,123,124,125,126,127].

Another vaccine that has been associated with thrombosis is the Ad26.COV2.S of Johnson & Johnson/Janssen [106,128,129,130,131,132]. Reported cases of thrombosis related to mRNA vaccines are rare [133,134,135,136,137,138,139,140,141,142].

In 2021, a detailed EMA analysis for 213 cases of CVT after vaccination against COVID-19 was published [143]. They reported 87.8% after vaccination with ChAdOx1 nCoV-19, 12.2% after BNT162b2 vaccination, and only one case after mRNA-1273 vaccination. Immediately after EMA, many other regulatory agencies also published the data from their surveillance systems for observed thrombotic complications after vaccination against COVID-19 [96,144,145,146,147,148,149,150,151,152].

Regarding thrombocytopenia, the UK reported 60 cases after the ChAdOx1 nCoV-19 vaccine and 34 cases with the BNT162b2 vaccine [58]. One hundred ninety-five cases have been reported to the USA Vaccine Adverse Event Reporting System (VAERS) following the use of BioNTech/Pfizer and Moderna vaccines [152]. 

Understanding the possible mechanisms underlying the coagulation pathways is crucial to managing thrombotic complications after vaccination against COVID-19. Careful and long-term surveillance of the administration of vaccines is necessary. It should not be forgotten that the frequency of severe thrombotic events/complications after vaccination is relatively low. The benefits and risks of the currently approved vaccines must be evaluated, and the possibility of developing severe and long-term complications (long COVID-19) after infection with the SARS-CoV-2 virus must be considered.

Early diagnosis and treatment are crucial for VITT, including high-dose intravenous immunoglobulins and anticoagulation [153]. 

An international network cohort study of four countries by Markus et al. characterized the treatment options of thromboembolic events following COVID-19 vaccination. Thrombosis with thrombocytopenia syndrome (TTS) has been identified as a rare adverse event following some COVID-19 vaccines. Various guidelines have been issued on the treatment of TTS. We aimed to characterize the treatment of TTS and other thromboembolic events (venous thromboembolism (VTE) and arterial thromboembolism (ATE). Most patients with TTS received heparins, platelet aggregation inhibitors, or direct Xa inhibitors. Most VTE patients (before and after vaccination) were first treated with heparins in inpatient settings and direct Xa inhibitors in outpatient settings. In ATE patients, treatments were similar before and after vaccinations, with platelet aggregation inhibitors prescribed most frequently. Inpatient and claims data also showed substantial heparin use. Heparin use in the treatment of post-vaccine TTS suggests that most events were not identified as vaccine-induced thrombosis with thrombocytopenia by the treating clinicians [154].

The management of these events was previously covered in the WHO Guidance for clinical case management of thrombosis with thrombocytopenia syndrome (TTS) following vaccination to prevent coronavirus disease (COVID-19) in 2021 [80]. We must acknowledge that vaccine-associated thrombosis could occur in individuals with or without thrombocytopenia. However, future studies should focus on identifying alternative thrombosis-promoting components to improve the safety and efficacy of future COVID-19 vaccines.

Last but not least, there are arterial thromboembolic incidents following the COVID-19 vaccination. Such a study was provided by Wills et al. [155]. The authors reported two stroke cases due to arterial and venous thromboses within four weeks following the administration of the AstraZeneca vaccine. The patients were both young and in good health overall, with a few risk indicators for thrombosis. However, they had a swift and eventually lethal decline in cerebral function. The individuals had a notable decrease in platelet count and an abnormally high level of D-dimers, both of which are frequently documented in this particular medical condition. In all instances, quantifiable immunoglobulin G platelet factor-4 antibodies were identified using the enzyme-linked immunosorbent test, resembling the antibodies observed in cases of heparin-induced thrombocytopenia [155].

Nevertheless, instances of arterial thrombotic episodes have also been observed, as evidenced by the cases presented. Laboratory investigations commonly demonstrate a decreased platelet count, along with significantly elevated D-dimer levels and decreased fibrinogen levels. The presence of antibodies against platelet factor 4 (PF4) has been detected in affected individuals, indicating parallels to heparin-induced thrombocytopenia (HIT) even without prior heparin exposure [155]. Another cerebral incident was presented by Berlot et al. involving a patient with complete arterial occlusion of the right internal carotid artery and middle cerebral artery 9 days after the first dose of AstraZeneca (ChAdOx1 nCov-19) vaccination [156]. 

Going further down the road, post-COVID-19 vaccination was also associated with retinal artery occlusions, with hypercoagulability and blood thrombotic disorders being the main risk factors [157]. However, there are still no official recommendations on the use of particular types of vaccines depending on the underlying risk factors in the individuals [158]. Usually, the timing of vaccination is the decisive factor—to schedule vaccination after controlling all the risk factors and underlying conditions [159]. 

Furthermore, there was a case report of a patient who experienced acute myocardial infarction within 24 h after receiving the initial dose of the mRNA-1273 vaccination. The occurrence of subacute in-stent thrombosis was observed several days following percutaneous coronary intervention. VITT was suspected based on identifying antiplatelet factor 4 antibodies in the patient’s serum. We present this patient case to raise awareness among physicians regarding the potential occurrence of VITT associated with mRNA-1273. Nevertheless, it is essential to note that these findings should not, in any manner, lower the level of enthusiasm surrounding vaccination [160]. 

Another illustrative case report was about a male 60-year-old patient who underwent administration of the initial dosage of the AstraZeneca vaccine. Subsequently, the patient sought medical attention, reporting symptoms of discomfort and abnormal sensations in the left hand. The presence of upper limb ischemia was identified using investigative measures [161]. 

## 10. Diagnostics of Coagulopathies following COVID-19 Vaccination through Imaging

Radiology plays a crucial role in the diagnosis of vascular diseases. Different diagnostic methods depend on the localization and type of the pathology. The methods are divided into two main groups—those that use ionizing radiation and those that do not [162].

In recent years, ultrasound diagnostics has undergone serious progress and has established itself as a leading method for diagnosing vascular diseases. In addition to offering diagnosis without ionizing radiation, ultrasound machines are widely available, and the examination cost is relatively low. The method is successfully used to diagnose large blood vessels, peripheral vessels, vessels of various organs and systems, and brain vessels. 

There are also various combinations of ultrasound and other methods, known as Fused Ultrasound. The method is available in combination with previously performed magnetic resonance imaging or computed tomography. Fused ultrasound is used to analyze prostate, liver, and breast diseases, and in neurological practice [163]. 

Computed tomography (CT) has rapidly developed in recent years with the introduction of multidetector computed tomography and dual-energy CT (DECT). A CT scan of the blood vessels is known as CT angiography (CTA) and CT venography (CTV). The method is widely available and non-invasive, and thanks to the latest software technologies, it is relatively easy to use [164]. In recent years, CTA and CTV have established themselves as the gold standard for diagnosing various blood vessel diseases—stenoses, thrombosis, aneurysms, dissections, obstructions, and bleeding, especially in emergencies. In recent years, the use of CT coronary angiography as a non-invasive alternative to classical coronary angiography has been imposed to diagnose coronary vessel diseases [165].

Recently, we documented nine case studies of anticoagulated COVID-19 patients with retroperitoneal and abdominal bleeding, emphasizing the power of CT to diagnose these complications early [166].

Magnetic resonance imaging is a modern, non-invasive method that also has a role in diagnosing blood vessels. The advantage of the method is that it can represent the vessel without using contrast material, which is particularly suitable for patients with impaired renal function, in which performing computer tomographic angiography is contraindicated. In addition to the native scan, it can also perform magnetic resonance angiography and magnetic resonance venography by applying contrast material. The contrast used in these studies is at a lower concentration than the iodine contrast used in computed tomography and can be used in patients with moderate renal function impairment. The informativeness of MPA and MPV is high, similar to that of CTA and CTV. The disadvantages of the method are the well-known disadvantages of magnetic resonance imaging in general—high cost, long-term examination, the presence of some metal implants and pacemakers, and, in most countries, no possibility of use in emergencies [167]. 

Thus, imaging could be critical in identifying and elucidating multiorgan complications, even in the rare case of SARS-CoV-2 vaccination [168,169]. Therefore, rapid recognition of complications following COVID-19 vaccination and appropriate treatment are appreciated.

## 11. Global Benefit/Risk Ratio of the Various Vaccination Types for Bleeding or Thrombotic Complications

Despite the reported cases in the literature, no clear causal relationship between vaccination and the pathology developed can be concluded with the current data. The European Medicines Agency (EMA) confirms that the overall benefit/risk ratio of COVID-19 vaccination remains positive. Due to reported menstrual bleeding after vaccination [170,171], the EMA has recommended that heavy menstrual bleeding be added to the product information as a side effect of unknown frequency for the mRNA vaccines. However, the available data reviewed involved mostly non-serious and temporary cases [172].

Our narrative review has some limitations. First, the topic of hereditary and acquired coagulation disorders is broad, and we could not cover all the disorders in vast detail. However, there is no evidence in the literature regarding some rare blood disorders and COVID-19 and COVID-19 vaccines. Second, we could not cover all the case reports covering bleeding and thrombotic events following COVID-19 vaccination, although we aimed to include the most significant. Third, there are no data on the booster doses and the risk of adverse events associated with coagulation. Therefore, we did not focus on this matter.

Along with these limitations, we believe that our paper has some strengths: this is the first review to comprehensively focus on both hereditary and acquired coagulation disorders in the light of COVID-19 and also to discuss the most-commonly described adverse effects of COVID-19 vaccines associated with bleeding or thrombotic events.

We want to emphasize that we discussed these events following COVID-19 vaccination. Still, there is no evidence of direct causality between the bleeding and thrombotic events and COVID-19 vaccines, although some hypotheses for immunological mechanisms were proposed. Based on such a hypothesis for the production of autoantibodies after vaccination, one should accept that the second and following doses will lead to immediate consequences in subjects who had already activated an abnormal immune response. However, no data support this in clinical or real-world settings [85].

Also, the prevailing data on the safety of COVID-19 vaccines confirm that the efficacy of vaccines against COVID-19 and their benefits to reduce morbidity, complications, and mortality outweigh the potential risk of rare serious adverse effects such as coagulation disorders [173,174]. The causal relationship could not be supported by analyzing data from the pharmacovigilance reports since their data are administrative and non-controlled. Thus, there is more of a statistical link than a correlation between adverse events and vaccination [175,176,177]. 

Regrettably, some people are still hesitant to acknowledge the risks posed by SARS-CoV-2, comparing them to earlier influenza outbreaks and ignoring the reality of mortality rates, and they hesitate to be vaccinated against COVID-19. The phenomenon of denial is significant, and reports of adverse effects, particularly thrombotic ones, have a bearing on it. A complicated phenomenon, vaccination hesitancy is influenced by people’s opinions of the effectiveness and safety of immunizations; therefore, more studies should focus on the safety and more studies should investigate vaccine fear in people [178,179,180].

Nevertheless, the incidence of thrombotic events is extremely low (1/100,000–1/1,000,000 vaccinated subjects) [107,146,181,182,183,184]. This is also valid for acquired hemophilia [185]. Furthermore, the rate of acquired hemophilia post-vaccination does not differ significantly from that of acquired hemophilia without vaccination, which has a basic prevalence of 1.5 per million, which does not allow us to estimate the hazard ratio for this condition [186].

Based on available data, Abrignani et al. [187] and Elalamy et al. [58] propose to avoid these unnecessary evaluations in the case of COVID-19 vaccination: systematic premedications with low molecular weight heparin, direct anticoagulants or aspirin, routine screening for thrombophilia, periodic assessment of PF4 antibodies after vaccination or monitoring changes in D-dimer, and systematic use of venous echo-Doppler examinations after the vaccine administration. 

Also, we must consider the possibility that previous infections with SARS-CoV-2 or other viruses may increase the risk of thrombotic events or inflammatory reactions following vaccination or other interventions [188]. However, careful surveillance and long-term follow-up of the safety and effectiveness of COVID-19 vaccines are still needed, especially to gain data on vaccine-vaccine interactions and use in immunocompromised and comorbid patients [150,178].

## 12. Conclusions

It should be kept in mind that severe thrombotic episodes seem to occur seldom following COVID-19 vaccination. Also, the prevailing data on the safety of COVID-19 vaccines confirm that the efficacy of vaccines against COVID-19 and their benefits to reduce morbidity, complications, and mortality outweighs the potential risk of vaccines for rare serious adverse effects, such as coagulation disorders. However, careful surveillance and long-term follow-up of the safety and effectiveness of COVID-19 vaccines are still needed, especially to gain data on vaccine–vaccine interactions and use in immunocompromised and comorbid patients.

## Figures and Tables

**Figure 1 biomedicines-11-02813-f001:**
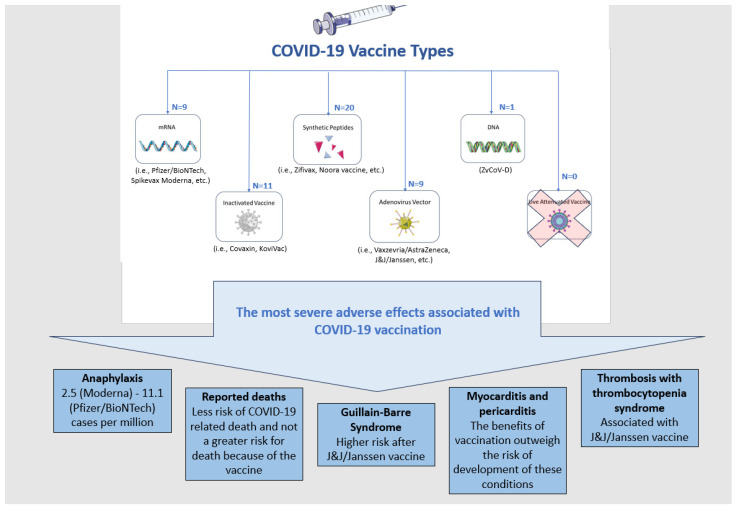
The approved COVID-19 vaccines and the most severe adverse events, although very rare, associated with their use.

**Figure 2 biomedicines-11-02813-f002:**
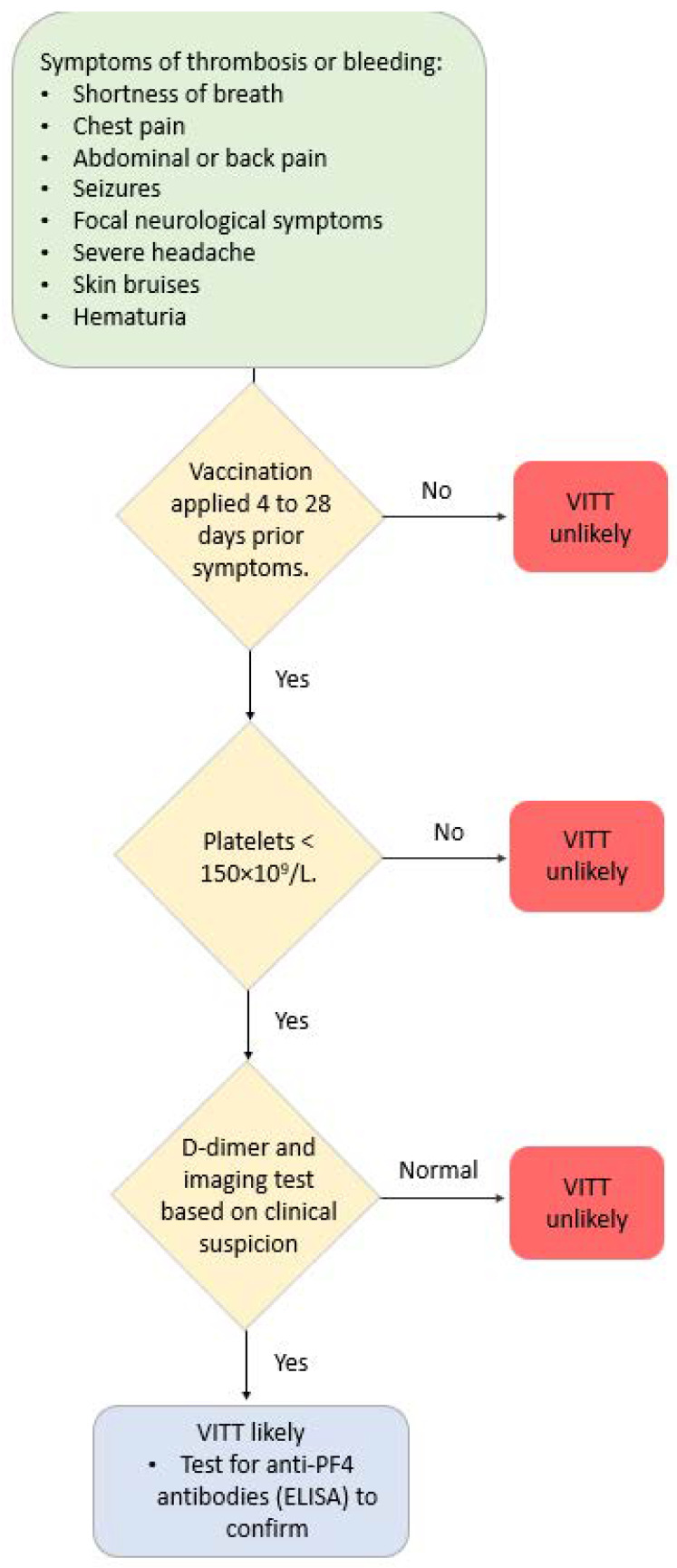
Diagnostic algorithm in patients with suspected VITT.

**Table 1 biomedicines-11-02813-t001:** VITT incidents following COVID-19 vaccination based on the data in the literature.

Type of Study	Vaccine	Number of Participants	Standardized Morbidity Ratio for Any Thrombocytopenia/Coagulation Disorders	Mortality	Authors
Population-based	Viral vectored	281,264	1.52 (0.97 to 2.25)	NR	Pottegård et al. [95]
European database	Viral vectored and m-RNA	NR	Viral vectored vaccine recipients 2.5 (2.3–2.7)mRNA vaccine recipients 1.9 (1.6–2.2)	0.4 and 4.8 deaths/1 million doses for mRNA and viral vector recipients, respectively	Cari et al. [96]
Systematic review	Viral vectored	4552 participants (12 studies)	OR 13.8; 95% CI 2.0–97.3	Approximately 1/3 were deceased	Palaiodimou et al. [97]
Systematic review	Viral vectored and m-RNA	69 participants (59 studies)	63 out of 69 (91.3%) occurred after the viral vector vaccine	24 deceased	Hafeez et al. [98]
Surveillance	m-RNA	10,162,227 (8 participating US health plans)	Excess cases in risk interval 1.0 (−4.6 to 1.4) per million doses	NR	Klein et al. [99]
Systematic review	Viral vectored and m-RNA	49 participants (12 studies)	NR	39% deceased	Sharifian-Dorche et al. [86]
Case series	Viral vectored and m-RNA	11 patients	NR	6 deceased	Greinacher et al. [85]
Case series	Viral vectored	39 cases	NR	51% deceased	Mouta Nunes de Oliveira et al. [100]

NR—not reported.

## Data Availability

Not applicable

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
