# Peer review of "Blood Coagulation and Thrombotic Disorders following SARS-CoV-2 Infection and COVID-19 Vaccination"

_biomedicines, 2023, doi:10.3390/biomedicines11102813_

Round 1

Reviewer 1 Report

This review extensively reports about coagulation complications of COVID19 vaccines. The length of the text and the reference list may be shortened and focussed on the main topics related to COVID19 vaccination:

1. Vaccine induced thrombosis with and without thrombocytopenia

2. Use of different vaccine types in patients depending from underlying risk factors

3. Rapid recognition of vaccine induced complications and appropriate treatment

The use of graphical based algorithms may improve the quality of the paper.

Author Response

Reviewer 1

Comments and Suggestions for Authors

This review extensively reports about coagulation complications of COVID19 vaccines. The length of the text and the reference list may be shortened and focussed on the main topics related to COVID19 vaccination:

Authors` reply:

- Thank you for your time to review our paper and for the overall evaluation of it as good.

- We acknowledge that our paper might have some issues in conformity with the referees` comments. We have addressed them and revised the manuscript accordingly. Changes are visible as highlighted and/or track changes. We revised the length of the manuscript, while preserving the journal requirements for the minimal word count. We also revised again the references, after reducing the unnecessary text (i.e., physiology of coagulation cascade). However, after adding more information suggested by the three reviewers, we add about 10 references. We would like to ask the reputable referee to accept this amount of references as necessary regarding the topic and the need of being accurate, comprehensive and profound .

  1. Vaccine induced thrombosis with and without thrombocytopenia
  2. Use of different vaccine types in patients depending from underlying risk factors
  3. Rapid recognition of vaccine induced complications and appropriate treatment

 Author’s reply: Thank you for your comment. We can agree that vaccine-associated thrombosis can occur with or without concomitant thrombocytopenia. We commented also on use of particular type of vaccines depending on the underlying risk factors in the individuals. The suggestions have been applied to the manuscript and the text and reference list have been shortened.

The use of graphical based algorithms may improve the quality of the paper.

Author’s reply: Thank you for your useful suggestion. The following algorithm for management of patients with suspected VITT has been included in the manuscript as Figure 2 in the manuscript.

Reviewer 2 Report

This article is first a kind of "melting pot" of scientific information on Covid-19/Sars-Cov-2, hemostasis physiopathology, bleeding disorders and thrombotic diseases, with some inputs concerning the major vaccine associated risk, which is VITT (Vaccine Induced Thrombotic Thrombocytopenia). The content matches only very poorly with the very attractive title, and only the VITT complications are discussed, and truely associated to the Covid-19 adenovirus-vector vaccines, but this is a minor part of that review. In addition, an extensive revision of the English syntax and wording is required. Especially, the section concerning the presentation of the various vaccines has a very poor syntax.

I think that this review should be extensively revised and structured differently by analyzing which are the associations of bleeding episodes or thrombotic diseases with the Covid-19 vaccines themselves. Of course, the major concern will focus on the ultra-rare VITT occurring after the vaccination with the adenovirus-vector vaccines, and the authors could extend the discussion to the possible risk associated with the use of the adenovirus vector for other clinical applications like gene therapy. Then, they could review the other milder side effects associated with the various vaccines, focusing on what is demonstrated and confirmed.

English is often approximative and sometimes unexpected in a scientific report. An extensive revision of the language is required.

Author Response

Reviewer 2

This article is first a kind of "melting pot" of scientific information on Covid-19/Sars-Cov-2, hemostasis physiopathology, bleeding disorders and thrombotic diseases, with some inputs concerning the major vaccine associated risk, which is VITT (Vaccine Induced Thrombotic Thrombocytopenia). The content matches only very poorly with the very attractive title, and only the VITT complications are discussed, and truely associated to the Covid-19 adenovirus-vector vaccines, but this is a minor part of that review. In addition, an extensive revision of the English syntax and wording is required. Especially, the section concerning the presentation of the various vaccines has a very poor syntax.

I think that this review should be extensively revised and structured differently by analyzing which are the associations of bleeding episodes or thrombotic diseases with the Covid-19 vaccines themselves. Of course, the major concern will focus on the ultra-rare VITT occurring after the vaccination with the adenovirus-vector vaccines, and the authors could extend the discussion to the possible risk associated with the use of the adenovirus vector for other clinical applications like gene therapy. Then, they could review the other milder side effects associated with the various vaccines, focusing on what is demonstrated and confirmed.

Author’s reply: Thank you for your critical notes. We acknowledge that our paper might have some issues in conformity with the referees` comments. We did our best to gather this information in the paper and not disappoint and mislead the readers with the title.

We applied corrections to the manuscript and its structure. The following structure of the manuscript was adopted:

Firstly, we focused on the inherited blood coagulation disorders and then on the inherited thrombotic disorders, discussing the current recommendations regarding patients with these pathology entities and COVID-19 vaccination.

Afterwards, the review focuses on coagulation and thrombotic disorders from one side, and bleeding events from the other, related to COVID-19 vaccination.

Lastly, we discuss the recommended diagnostic and therapeutic interventions in patients with thrombotic or bleeding disorders following COVID-19 vaccination.

In order to convey the message of the review in a clearer way we added the two following figures: Figure 1. Illustrating the different types of vaccines and major severe adverse effects associated with COVID-19 vaccination; Figure 2. Algorithm for management of patients with suspected VITT.

Comments on the Quality of English Language

English is often approximative and sometimes unexpected in a scientific report. An extensive revision of the language is required.

Author’s reply: Thank you for your constructive comment. Extensive English revision has been performed.

Reviewer 3 Report

In the review article by Dr Sekulovski et al., the Authors reviewed existing scientific literature on  the blood coagulation and thrombotic disorders followed by vaccination against COVID-19. In my opinion, the review is thorough and interesting, however I have one major and a couple of minor comments.

1. Major comment

The Authors claimed that they focused on both thrombotic and bleeding disorders, however as regards thrombotic disorders, however very little has been written on arterial thrombosis in which a role of blood platelets is very important. There are quite a few papers on this. I would suggest to add the respective part in the review devoted to this issue.

Minor comments:

1. Lines 58-59 „leading to more than six million deaths globally”. The actual number of deaths is almost 7 milion, please use newer reference than this dated on 2021

(1. WHO. WHO COVID-19 dashboard - up to date data on pandemic. WHO Heal Emerg Dashboard 2021. 642 https://covid19.who.int/region/searo/country/id [Accessed on July 10, 2023])

2. Table 1. I would suggest to move a left-hand column (with references) to the far right side of the table.

3. At least one figure would be recommended to enrich the review.

English is OK.

Author Response

Reviewer 3

In the review article by Dr Sekulovski et al., the Authors reviewed existing scientific literature on  the blood coagulation and thrombotic disorders followed by vaccination against COVID-19. In my opinion, the review is thorough and interesting, however I have one major and a couple of minor comments.

Dear Reviewer, thank you  for your time and overall commitment  in reviewing our article.

  1. Major comment

The Authors claimed that they focused on both thrombotic and bleeding disorders, however as regards thrombotic disorders, however very little has been written on arterial thrombosis in which a role of blood platelets is very important. There are quite a few papers on this. I would suggest to add the respective part in the review devoted to this issue.

We appreciate your key-point note. We have adopt this important scientific information in our article. Please find the information at the end of Chapter 9. (Thrombotic events following COVID-19 vaccination: hypothesis for pathogenesis and therapeutic interventions)

Minor comments:

  1. Lines 58-59 „leading to more than six million deaths globally”. The actual number of deaths is almost 7 milion, please use newer reference than this dated on 2021

(1. WHO. WHO COVID-19 dashboard - up to date data on pandemic. WHO Heal Emerg Dashboard 2021. 642 https://covid19.who.int/region/searo/country/id [Accessed on July 10, 2023])

Thank you for your valuable and important note. We have corrected the actual  data according to the current WHO Dashboard.

  1. Table 1. I would suggest to move a left-hand column (with references) to the far right side of the table.

Thank you for your note. We have corrected the table as you suggested.

  1. At least one figure would be recommended to enrich the review.

Author’s reply: Thank you for your suggestion.

A figure illustrating the different types of COVID-19 vaccines has been added as Figure 1, and also we added an algorithm for managing VITT.

Comments on the Quality of English Language

English is OK.

  • Thank you for the note.

Round 2

Reviewer 2 Report

Global comments:

Many thanks to the authors for their extensive revision and resubmission of the manuscript, which is improved for some of the aspects. Introduction of figures 1 and 2 is useful for the global reading and understanding. 

However, I still have some concerns with the global presentation and organization of this ointeresting review. The title should modified and be better like "Blood coagulation and thrombotic disorders following Covid-19 disease and vaccination" to better reflect the review content.

Then, I suggest that the authors slightly modify the review structure as follows:

First, review and summary of congenital (better than inherited; some are not hereditary but appear congenitally) coagulation and thrombotic  disorders.

Review of acquired coagulation and thrombotic disorders.

Bleeding and thrombotic (mainly) complications developped during or after the Covid-19 course.

Covid-19 vaccination and occurrence of bleeding or thrombotic complications (to be classified by vaccine type), and statistical diffrences (if any?) with a matched non-vaccinated group.

Special case of the ultra-rare VITT complications occurring only with the adenovirus vector vaccine.

Global benefit/risk ratio of the various vaccination types for the bleeding or thrombotic complications.

Discussion and conclusions

Minor comments:

English language and syntax requires still attention.

On figure 1, it should be useful to mention the various vaccine brands in the boxes reporting the vaccine types (as the brand names are already indicated for the side effect boxes (figure bottom).

Why do authors associate anticoagulant therapy to reported bleeding events like in lines 344-346? The major complication of Covid-19 is first thrombo-inflammation and thrombosis, and bleeding could be the result of DIC evolution.

Why do the authors name paragraph 7 "Bleeding events after Covid-19 vaccination"? This paragraph mainly concerns thrombocytopenia and VITT. The life-threatening complications developed are nor "hemorrhagic" but "thrombotic trhombocytopenia", with sometimes DIC (and therefore some bleeding, but only as the consequence of exhacerbated fibrinolysis following disseminated thrombosis).

English language and syntax must be revised and improved.

Author Response

Reviewer 2

Global comments:

Many thanks to the authors for their extensive revision and resubmission of the manuscript, which is improved for some of the aspects. Introduction of figures 1 and 2 is useful for the global reading and understanding. 

However, I still have some concerns with the global presentation and organization of this interesting review. The title should modify and be better like "Blood coagulation and thrombotic disorders following Covid-19 disease and vaccination" to better reflect the review content.

  • Dear reviewer, thank you for your valuable feedback on our manuscript titled "Blood Coagulation and Thrombotic Disorders Following Covid-19 Disease and Vaccination." We appreciate your careful review and insightful suggestions for improvement. We have addressed your comments and concerns as follows.
  • Title Modification: We agree with your suggestion to modify the title to better reflect the content of the review. The revised title is now: "Blood coagulation and thrombotic disorders following COVID-19 disease and vaccination." We believe this title more accurately captures the scope of the manuscript.
  • However, we are a bit concern about the title because “disease” part of COVID-19 abbreviation and would kindly ask the reviewer for this title: Blood Coagulation and Thrombotic Disorders Following SARS-CoV-2 infection and COVID-19 Vaccination.

Then, I suggest that the authors slightly modify the review structure as follows:

First, review and summary of congenital (better than inherited; some are not hereditary but appear congenitally) coagulation and thrombotic  disorders.

Review of acquired coagulation and thrombotic disorders.

Bleeding and thrombotic (mainly) complications developped during or after the Covid-19 course.

Covid-19 vaccination and occurrence of bleeding or thrombotic complications (to be classified by vaccine type), and statistical diffrences (if any?) with a matched non-vaccinated group.

Special case of the ultra-rare VITT complications occurring only with the adenovirus vector vaccine.

Global benefit/risk ratio of the various vaccination types for the bleeding or thrombotic complications.

Discussion and conclusions

  • Thank you for your valuable remarks. We completely agree with you that our manuscript structure needed some improvements. Hence, we have followed the proposed from you structure and modified the manuscript. All of our revisions can be tracked with Track changes.

Minor comments:

English language and syntax requires still attention.

  • We have carefully reviewed the English language and syntax, making necessary improvements to enhance readability and clarity.

On figure 1, it should be useful to mention the various vaccine brands in the boxes reporting the vaccine types (as the brand names are already indicated for the side effect boxes (figure bottom).

  • In Figure 1, we have added the names of various vaccine brands in the boxes, aligning with the side effect boxes as you suggested.

Why do authors associate anticoagulant therapy to reported bleeding events like in lines 344-346? The major complication of Covid-19 is first thrombo-inflammation and thrombosis, and bleeding could be the result of DIC evolution.

  • Regarding the association of anticoagulant therapy with reported bleeding events (new lines 277-279 and 344-346 ), we have clarified that bleeding may occur as a result of DIC evolution, which is often a consequence of thrombo-inflammation and thrombosis in Covid-19.

Why do the authors name paragraph 7 "Bleeding events after Covid-19 vaccination"? This paragraph mainly concerns thrombocytopenia and VITT. The life-threatening complications developed are nor "hemorrhagic" but "thrombotic trhombocytopenia", with sometimes DIC (and therefore some bleeding, but only as the consequence of exhacerbated fibrinolysis following disseminated thrombosis).

  • We agree with the reviewer and rearrange the passages and titles accordingly to accurately represent its content, focusing on thrombocytopenia and VITT complications, which can lead to thrombotic thrombocytopenia with DIC and, occasionally, bleeding as a consequence.

Comments on the Quality of English Language

English language and syntax must be revised and improved.

  • We believe these revisions address your concerns and improve the overall quality and organization of the manuscript. We appreciate your guidance and are committed to delivering a high-quality review article. If you have any additional comments or suggestions, please feel free to let us know.
  • Thank you once again for your thoughtful review.

Reviewer 3 Report

The Authors have modified the article according to my suggestions. I have no further comments.

Author Response

  • We believe these revisions address your concerns and improve the overall quality and organization of the manuscript. We appreciate your guidance and are committed to delivering a high-quality review article. 
  • Thank you once again for your thoughtful review.

Round 3

Reviewer 2 Report

Thank you for the extensive changes introduced in the text, which render it easier to read. This complies better with the expected presentation for this type of review.

The authors new proposal for the titer is fully acceptable. Thank you.

Some editing of English language i still required.

The revised version is better acceptable. Some editing of the English language is recommended.